# Quality of Life in Children and Adolescents with Stickler Syndrome in Spain

**DOI:** 10.3390/children9081255

**Published:** 2022-08-19

**Authors:** Juan José Fernández-Pérez, Paloma Mascaraque-Ruiz, Carlos Martín Gómez, Ignacio Martínez-Caballero, Teresa Otón, Loreto Carmona, Sergio Lerma Lara

**Affiliations:** 1Grupo de Investigación en Fisioterapia Toledo (GIFTO), Facultad de Fisioterapia y Enfermería, Universidad Castilla-La Mancha, 45071 Toledo, Spain; 2Facultad de Ciencias de la Salud, CSEU La Salle, UAM, 28023 Madrid, Spain; 3Hospital Infantil Universitario Niño Jesús, 28009 Madrid, Spain; 4Instituto de Salud Musculoesquelética, 28045 Madrid, Spain

**Keywords:** Stickler syndrome, quality of life, psychosocial factor, prevalence, paediatric

## Abstract

Objective: To describe the quality of life and daily functioning of Spanish children and adolescents living with Stickler syndrome (SS) and to estimate the prevalence of associated disease features in a representative sample. Methods: A cross-sectional study of children and adolescents with SS were recruited via telephone calls through the Spanish SS Association. All participants underwent a structured clinical interview and filled in questionnaires reporting their quality of life (EuroQol-5D, TSK-11, CHAQ and PedsQoL). The prevalence of the main features associated with the syndrome and the mean scores of the questionnaires were estimated with 95% confidence intervals (95% CI). Results: The recruited sample included 26 persons who were mainly children (mean age 10.4 ± 4.5 (SD) range: 5–14) and male (65.4%). The prevalence estimates of SS features were as follows: the presence of moderate pain (52%), hearing loss 67% (95% CI: 54.8 to 91.3) and myopia 96% (95% CI: 87.2 to 104.4). The mean scores of the QoL indices were as follows: 22.4 (95% CI: 19.2 to 25.5) (±7.5) for TSK-11; 76.2 (95% CI: 68.8 to 83.6) (±17.1) for PedsQoL, 0.8 (95% CI: 0.7 to 0.9) (±0.3) for EQ-5D and 0.61 (95% CI: 0.24 to 1.0) (±0.9) for the cHAQ functional index. Conclusions: Our results confirmed a high variability in syndrome-related manifestations, with a large prevalence of visual and hearing deficits, pain and maxillofacial alterations. These findings may facilitate the detection of the most prevalent problems in this population, which could be a target to be addressed during the treatment of children and adolescents with SS.

## 1. Introduction

In 1965, Stickler et al. described a syndrome characterized by megalophthalmos, retinal detachment, deafness, cleft palate, Pierre Robin sequence, joint hypermobility and premature arthritis [1,2]. Stickler syndrome (SS) is a dominantly inherited connective tissue disorder of fibrillary collagen, with high variability in the manifestation of phenotypes, both between and within families [3,4]. Genetically, SS is heterogeneous, with responsible mutations described in the COL2A1, COL11A1 [5] and COL11A2 procollagen genes, leading to various degrees of the abnormal synthesis of collagen types II, XI or IX. It is considered a rare disease, with prevalence figures ranging from 1 to 9 per 10,000 people and an approximate incidence rate of 1 in 7500 to 9000 newborns [6,7].

Most of the musculoskeletal manifestations of this syndrome are treated surgically, especially those related to the Pierre Robin sequence, which can cause airway obstruction and feeding disturbances [8]. Other common bone manifestations, such as scoliosis [9] or hip abnormalities [10,11], are usually treated invasively, with some of these patients undergoing at least one surgery before the age of 18 [9]. Despite the musculoskeletal findings present in this population, it is noteworthy that no conservative rehabilitation-based treatments have been reported in the literature for these patients. Regarding other usual problems, a total of 40–80% of patients with SS develop retinal detachments and tears. However, these alterations are corrected using prophylactic laser treatment, which was shown to be effective in the treatment of these disorders [12].

Patient-reported outcomes (PROs), such as self-esteem, psychosocial functioning and health-related quality of life (HRQoL), are essential for obtaining more detailed information about a patient’s health. This is because HRQoL is a very broad concept, encompassing not only physical health but also the social and psychological state of individuals. It is for this reason that PROs related to the quality of life provide real information about the patient’s view of their health from a more global perspective. The HRQoL measures are well documented in the literature for adults and children [13]. Children with collagenopathies usually experience mild complex pain, which negatively impacts their quality of life [14]. Moreover, these children also present variability in clinical and musculoskeletal manifestations, as well as alterations in bone formation, which cause difficulties in participation in daily life activities [15].

Nevertheless, these measures have seldom been explored in patients with SS. Patients with SS require frequent outpatient clinic visits and usually need to be admitted to hospitals due to surgery, complications or special treatments, all of which have an impact on daily life. In addition, physical abnormalities, especially in the face, are very typical of this syndrome and are negatively correlated with HRQoL and behavior [16,17]. Furthermore, studies that summarize the clinical manifestation and psychosocial factors in these children are necessary to obtain an early diagnosis [18] and a better understanding of factors that may influence the development of this population, which may be clinically relevant for future treatment goals related to bone disorders [19], pain and, consequently, loss of quality of life. To our knowledge, no studies on children and adolescents measured the impact of SS on their quality of life.

For this reason, the aim of the present study was to describe the HrQoL of Spanish children and adolescents living with SS and to estimate the prevalence of the different features of SS.

## 2. Materials and Methods

A cross-sectional study was conducted. The study protocol and materials were approved by the ethics committee of “Centro Superior de Estudios Universitarios La Salle” (CSEULS) in Madrid, Spain.

Patients were eligible for the study if they had a medical diagnosis of SS, their age was between 4 and 18 years and they had the ability to walk at least eight meters. Eligible participants were recruited from December 2017 to March 2018 by calling those on the Spanish Stickler’s Syndrome Association roll.

Patients and their families were received into La Salle Motion-Lab (M-Lab), signed an informed consent form and were interviewed by one of the researchers. Quality of life was assessed with a series of PROs: (1) The EuroQol 5D-3L, which is a widely used self-administered questionnaire that covers 5 domains (mobility, self-care, activities of daily living, pain/discomfort and anxiety/depression), which are rated from not affected to maximum impact and include a general health visual analogue scale (VAS) from 0 “poorest health” to 100 “perfect health” [20], which is a valid measure for the Spanish population [21]. (2) The Tampa Scale of Kinesiophobia-11 items (TSK-11), which is the short version of the Tampa Scale for Kinesiophobia and is one of the most used scales for measuring pain-related fear in patients with musculoskeletal chronic pain; scores range from 11 to 44, with larger scores representing a larger amount of fear [22]. This scale showed good reliability (Cronbach’s α = 0.79) and good validity compared with other scales for measuring psychosocial factors for persistent pain [22]. (3) The Pediatric Quality of Life Inventory (PedsQL 3.0 rheumatology module), which is a 20-item questionnaire with 5 domains (pain and aches, daily activities, treatments, concerns and communication); the scores range from 0 to 100, with higher scores indicating a better quality of life [23,24,25]. PedsQL demonstrates high reliability (Cronbach’s α = 0.83) for caregivers in the Spanish version [26]. In addition, daily functioning was measured with the Child Health Assessment Questionnaire (CHAQ), which is a validated questionnaire for children with juvenile idiopathic arthritis and validated in other rheumatic diseases. This scale ranges from 0 to 3, with the highest score indicating the inability to perform activities [27,28]; the Spanish translation is a valid scale for children with juvenile idiopathic arthritis [29]. According to the literature, the time needed to complete each questionnaire was a few minutes for EuroQoL-5D [30], 3 min for TSK-11 [31], 10 min for PedsQL [32] and 5 min for CHAQ [33], which is similar to our experience.

The statistical analysis consisted of a description of the variables using summary statistics (mean and standard deviation (SD) or median and interquartile range (IQR) for numerical ratings and absolute and relative frequencies for categorical variables). In addition, the confidence intervals for means and prevalences were estimated. The analysis was done with Stata v 12 statistical software (StataCorp LLC, College Station, TX, USA).

## 3. Results

The sample included 26 subjects, 17 (65%) of whom were male and 9 were female, with a mean age (SD) of 10.4 (4.5) years. This sample comprised half the population of members of the Spanish SS association. The socio-demographic, family background and other perinatal characteristics are presented in Table 1. Visual disturbances and musculoskeletal disorders were frequent in the family background, and swallowing disorders and hypotonia were frequent at birth.

The spectrum of clinical manifestations is shown in Table 2. The most frequent manifestations were a round face with 18 (72%), a flat nasal root with 16 (67%) and hearing loss with 19 (73%). Nearly all patients (23, 96%) had myopia.

Table 3 summarizes the data on the different PROs in actual results and estimates (with 95% CIs). The results of the EuroQoL-5D are presented as a percentage of children and adolescents in each of the levels of concomitance (from lower to higher impact), as the calculated score and as the VAS.

## 4. Discussion

The spectrum of clinical manifestations found in this national sample of Spanish children and adolescents with SS was diverse, and the scores for the PROs were limited to different extents, depending on the domain measured. Moreover, this was the first study to compile data on the quality of life in children with SS. Other studies in the literature made classifications based on genetics and radiological findings in SS [34,35,36], leaving aside the analysis of psychosocial factors and quality of life.

PROs offer insight into the subjective judgments of patients and their families and may have a significant weight in treatment decisions [37]. In our study, there was high variability in the impact of health on aspects of daily life, depending on the domain explored. For instance, 52% of children reported some pain and a moderate level of kinesiophobia, reflecting a relative fear of movement and towards hurting oneself. However, the levels of functioning and quality of life, despite being affected, were better than expected [38]. Nevertheless, these factors can lead to long-term pain, as was shown with other pathologies, such as juvenile idiopathic arthritis, where 1 in 10 had recurrent pain-related problems [39]. Indeed, early identification of fear movement may be beneficial for the recovery of patients due to the association between kinesiophobia and persistent pain [38]. For this reason, it is important to be aware of the characteristics of these children to prevent future sequela as pain. In this sense, our results showed a very good opportunity for rehabilitation in these children and adolescents.

Quality of life assessment is key to understanding a patient’s perception of the disease and its consequences on daily activities. Interestingly, these results showed a relatively low impact on functional capacity or quality of life. Furthermore, the percentages of musculoskeletal problems were relatively low (e.g., 17.4% with osteoarthritis and 30% with hypermobility syndrome), while those affecting sight and hearing had a greater presence (e.g., 73% with hearing loss and 96% with myopia). Moreover, the nurturing from their families and small associations like the one these patients belonged to may have a large impact on perception. However, other aspects, such as interaction with peers and self-esteem, which are critical at these ages, were not explored. Hong et al. performed a study focusing on HRQoL in patients with syndromic and isolated Robin sequence that was treated with mandibular distraction osteogenesis or tracheostomy [40]. The results showed no difference in the HRQoL between syndromic Robin sequence and isolated Robin sequence patients, nor between the tracheostomy and mandibular distraction osteogenesis; overall, all patients were affected. There are several studies in the literature that describe the HRQoL in children with collagenopathies, such as osteogenesis imperfecta. The study conducted by Y. Song et al. (2018) [41] and Vanz A. et al. (2018) [42] used the PedsQL scale to quantify the HRQoL in children with osteogenesis imperfecta. Their results for the mean punctuation on the PedsQL scale (65.6 ± 23.8 vs. 71.13 ± 12 vs. 76.2 ± 17) were slightly lower than those obtained in our study. This comparison showed that these SS children had similar HRQoLs as those with osteogenesis imperfecta.

Our descriptive study on QoL in children and adolescents with SS has clinical implications that should be highlighted. First, the presentation of clinical and morphological characteristics may facilitate early diagnosis, limiting and preventing possible future alterations that may occur during their development.

Second, this study showed that clinical manifestations present in children do not greatly affect their quality of life. Nonetheless, the presence of pain and kinesiophobia in a large percentage of these children could be perpetuated over time. This could cause their quality of life to be reduced as the disease progresses, which added to the high rate of surgical interventions and could be a major problem to be considered. In fact, the presence of musculoskeletal pain for more than 4 months in adolescents was associated with worse pain modulation [43] and pre-surgical pain intensity is associated with chronic post-surgical pain in children [44]. To improve the quality of life of these children, the first line of treatment should be based on conservative treatment using strength and endurance exercise programs. A systematic review of children with juvenile idiopathic arthritis found favorable results in increasing the quality of life of these children when performing exercise training programs 2–3 days a week for 3–6 months [45]. Hence, the characteristics described in this population may help to initiate new conservative treatment approaches for children and adolescents with SS. Future studies should consider a conservative treatment approach, such as exercise protocols for these patients, with the aim of improving their physical condition, and thus, their quality of life.

Nevertheless, our study was not without limitations. Our results might be influenced by the low number of participants and by an information bias. We only recruited children and adolescents from the national SS association, which further limited the number of subjects we were able to recruit. It should also be noted that some of the questionnaires were completed by parents; therefore, the results may not accurately reflect the quality of life and functional capacity of the children. Nevertheless, as parents “know their child best”, parental perceptions of health and HRQoL are important and informative regarding treatment decisions [46].

Overall, given the scarce information on SS, this work provides a starting point for the comprehension of aspects related to the quality of life and functional capacity in children and adolescents with SS.

## 5. Conclusions

In conclusion, our results confirmed a large prevalence of deficits in children and adolescents living with SS, with the most relevant being facial malformations, moderate pain, and ocular and auditory alterations, while only a mild reduction in quality of life was found.

## Figures and Tables

**Table 1 children-09-01255-t001:** Descriptive characteristics of the sample of children and adolescents with Stickler syndrome.

Variables	N *	Value
Sex, *n* (%)	26	
Male		17 (65.4)
Female		9 (34.6)
Age (years), m (SD)	26	10.4 (4.5)
median (IQR)		11 (5–14)
Weight (kg), m (SD)	25	40.2 (19.0)
median (IQR)		40.8 (21.4–55.9)
Height (m), m (SD)	25	1.4 (0.2)
median (IQR)		1.5 (1.2–1.6)
Family background, *n* (%)		
Visual disturbances	25	20 (80)
Hearing impairments	25	11 (44)
Musculoskeletal disorders	24	11 (46)
Cleft palate	24	4 (17)
Perinatal development, *n* (%)		
Problems during pregnancy	25	7 (28)
Swallowing problems at birth	25	12 (48)
Hypotonia at birth	24	8 (33)
Problems in psychomotor development <age 1	25	7 (28)

* Effective *n*, which was used as a denominator. Abbreviations: m, mean; SD, standard deviation; IQR, interquartile range.

**Table 2 children-09-01255-t002:** Prevalence estimates of Stickler syndrome’s features in children and adolescents of a national patients’ association.

Variables	N *	Value	95% Confidence Interval
Cleft palate	25	10 (40)	19.3 to 60.6
Micrognathia	25	9 (36)	15.7 to 56.2
Glossoptosis	25	4 (16)	0.5 to 31.4
Pierre Robin sequence	25	6 (24)	6 to 42.0
Round face	25	18 (72)	53.1 to 90.9
Malar hypoplasia	24	4 (17)	0.6 to 32.7
Ocular proptosis	25	5 (20)	3.1 to 36.8
Flat nasal root	24	16 (67)	46.3 to 87.0
Hearing loss	26	19 (73)	54.8 to 91.3
Myopia	24	23 (96)	87.2 to 104.4
Retinal detachment	23	8 (35)	13.7 to 55.8
Cataracts	24	5 (21)	3.3 to 38.5
Peripheral alterations in eye’s fundus	23	5 (22)	3.5 to 40.0
Vitreous alterations	23	8 (35)	13.7 to 55.8
Osteoarthritis	23	4 (17)	0.6 to 34.1
Hypermobility syndrome	23	7 (30)	10.1 to 50.7
Marfanoid habit	23	4 (17)	0.63 to 34.1
Spinal dysplasia	23	2 (9)	−3.8 to 21.1
Muscular atrophy	23	4 (17)	0.6 to 34.1

* Effective *n*, which was used as a denominator. All values correspond to *n* (%).

**Table 3 children-09-01255-t003:** Quality of life and functional indices estimates in children and adolescents with Stickler syndrome.

PRO	N *	Value	95% CI
EuroQol-5D	25		
Domains, *n* (%)			
Mobility	25		
Level 1		20 (80)	63.1 to 96.8
Level 2		4 (16)	0.55 to 31.4
Level 3		1 (4)	−4.2 to 12.2
Self-care	25		
Level 1		21 (84)	68.5 to 99.4
Level 2		4 (16)	0.5 to 31.4
Level 3		0	
Daily activities	25		
Level 1		19 (76)	58.0 to 94.0
Level 2		6 (24)	6.0 to 42.0
Level 3		0	
Pain	25		
Level 1		11 (44)	23.1 to 64.9
Level 2		13 (52)	30.9 to 73.0
Level 3		1 (4)	−4.2 to 12.2
Anxiety	25		
Level 1		20 (80)	63.1 to 96.8
Level 2		5 (20)	3.1 to 36.8
Level 3		0	
Score			
Mean (SD)		0.8 (0.3)	0.7 to 0.9
Median (IQR)		0.9 (0.8–1)	
VAS global health	24		
Mean (SD)		84.3 (15.2)	77.9 to 90.7
Median (IQR)		87.5 (77.5–97.5)	
Tampa Scale of Kinesiophobia-11 items	25		
Mean (SD)		22.4 (7.5)	19.2 to 25.5
Median (IQR)		23 (18–26)	
Pediatric Quality of Life Inventory	24		
Mean (SD)		76.2 (17.1)	68.8 to 83.6
Median (IQR)		76.8 (71.9–87.5)	
Child Health Assessment Questionnaire	26		
Mean (SD)		0.61 (0.9)	0.2 to 1.0
Median (IQR)		0.1 (0–1.25)	

* Effective *n*, which was used as a denominator. Abbreviations: PRO, patient-reported outcome; CI, confidence interval; SD, standard deviation; IQR, interquartile range; VAS, visual analogue scale.

## Data Availability

Not applicable.

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
