# Peer review of "Quality of Life in Children and Adolescents with Stickler Syndrome in Spain"

_children, 2022, doi:10.3390/children9081255_

Round 1
Reviewer 1 Report
The aim of the submitted work is to assess the quality of life and prevalence of different clinical signs of patients with Stickler syndrome in Spain with an age between four and eighteen years who have the ability to walk at least eight meters.
Collection of the parameters considered was done by administering questionnaires.
The sample consisted of 26 patients, of whom 17 males and 9 females.
Although it may be an interesting goal to assess quality of life in patients with Stickler syndrome, in my opinion, the limited number of patients enrolled for the study is the main limitation of the work.
The conculsions related to clinical features add nothing to what is already known while the quality of life assessments may not be adherent to reality even considering that, in some cases, due to the age of the patients, the answers to the questionnaires were provided by the parents.
I therefore suggest extending the sample by activating a national or international survey on the topic to ensure that the results are certainly adherent to the patients' conditions and thus can be a useful reference for professionals who support children with Stickler Syndrome.
I believe that the work, with such a limited case history, is not sufficient for publication
Reviewer 2 Report
I think the addressed issues are very interesting and suitable for Children. Indeed, specific studies about HRQoL of people with SS are lacking.
Also, I think the article is very improvable.
Here are some problematic points and suggestions:
- Title (and throughout the text): why do you use the word “youngsters” and not “adolescents”? In fact, a criterion of eligibility was “age between 4 and 18 years” (children and adolescents) and not 15-24 (range of the UN’s definition of “youth”).
- Abstract: no words are spent about Conclusions. At least one sentence summarizing the most important ones is needed.
- Introduction: the meaning of the following sentence [Despite their importance for optimal care and treatment success, patient-reported outcomes (PROs), such as self-esteem, psychosocial functioning, and health-related quality of life (HRQoL), are essential for obtaining more detailed information about the patient’s health] is obscure: please, clarify.
Aims: you underlined the aim «to analyze the impact of SS in the HRQoL of Spanish children and youngsters and to estimate the prevalence of the different features of SS», but “analyze the impact” implies an experimental design or employing a regression analysis. The Methods section as described here is not sufficient to bring out "the impact". A good example you can follow for the above aim is the Vanz et al.’s study (your [30] reference), that estimates the impact of the detected variables on PedsQL domains. I think your data allows for this analysis.
- Materials and Methods: several points are unclear. For example, are respondents the 26 patients or their familiars (parents or others?); is the used version of EQ-5D the Y version, suitable for children and teenagers? What version of PedsQL (child self-report or parent proxy-report) did you use? and the 4.0? What are the psychometric characteristics of the Spanish versions of the questionnaires? Are they similar to those of the original instruments? And so on.
- Results: In my opinion, reporting the ranges of the means and medians would make the results better readable. Why is the N denominator almost always different from 26?
- Discussion: this section reflects the limits of the other sections: why do you affirm that «resilience is usually very strong in children with rare diseases, and, despite limitations, they tend to function as well as healthy children»? A measure of resilience is lacking and is lacking a comparison with the not-SS children too.
I’m agree “the analysis of psychosocial factors and quality of life” is fundamental to improve the treatment procedures also in patients with Stickler Syndrome. And that your study is valuable in that sense, but it needed a deepened revision.
- References: no DOI is reported – why? Neither Varni (author of PedsQl) is cited! This section needs a revision too.
Well, the subject of article is very important; so, good work!
Reviewer 3 Report
Thank you very much for the chance to review this manuscript. The authors assessed the prevalence of disease features as well as the PROMs of patients with Stickler Syndrome. Overall, they recruited 26 patients.
First, I would like to thank the authors for their efforts in including the patients’ views on the disease. PROMs are often not considered in research, and I believe they are very important to evaluating treatment success. However, this might also be the bottleneck of this research. I cannot see how these PROMs provided by the authors can be translated into clinics. The authors did not assess treatment success, for example. So, the conclusion of the PROMs assessment is limited and, I think, not a very valuable contribution to this research field. However, I think assessing the prevalence of disease features is important as this might be used for diagnostic purposes, although only 26 patients were included. I would have felt better reviewing this manuscript if the authors had included a longer time frame for recruiting patients to increase the sample size, assessed other relevant factors (treatments), and compared the PROMs statistically to dig deeper into the question of how PROMs are affected.
Nevertheless, I would recommend publication after major revision, although I think this work is of limited impact. The following revision comments should be addressed before publication:
Major:
1)please include information in the introduction section on how the syndrome is treated after the etiology/epidemiology section (P1L29-37)
1)Please include a section in the discussion part discussing why your results are an important contribution to the field and what future studies can do better, building on your findings.
Minor:
1)Some typos need to be corrected (e.g., P2L58 “tthe”), and English should be polished by a native speaker
2) P2L89: “Stata v 12” please also provide city and country, e.g., StataCorp., College Station, TX, USA
Reviewer 4 Report
The Authors present a paper: " Quality of life in children and youngsters with Stickler Syndrome in Spain" that is a good paper. Introduction, Materials and Methods as well as figures and Tables are well reported. As indicated by Authors the reported cases are few but the disease is a very rare disease and the number is enough to draw some results. I have only minor sugestions for completing the manuscript:
1. please indicate the time necessary to fill out the questionnaires and the suggested timing to administer the questionnaires.
2. In the Conclusions please give some suggestion on how to improve from a quality of life point the detected deficits in these children
Round 2
Reviewer 2 Report
Dear Authors,
in my opinion the article has been improved a lot.
I don't agree with just one passage of the purpose: "the aim of the present study was to analyze the impact of SS in the HRQoL of Spanish children and adolescents .....". I think your research design highlights a concomitance and not the impact of the SS condition on HRQoL. In fact, HRQoL does not seem particularly compromised. A regression analysis could better clarify the "impact". Please, review the text of the aim and results highlighting the concomitance and not the impact.
Finally, you write: "Despite their importance for optimal care and treatment success, [...] patient-reported outcomes (PROs) are essential for obtaining more detailed information about the patient’s health". I'm agree, PROs are essential: why "despite their importance ...."? - I don't understand!
Author Response
I don't agree with just one passage of the purpose: "the aim of the present study was to analyze the impact of SS in the HRQoL of Spanish children and adolescents .....". I think your research design highlights a concomitance and not the impact of the SS condition on HRQoL. In fact, HRQoL does not seem particularly compromised. A regression analysis could better clarify the "impact". Please, review the text of the aim and results highlighting the concomitance and not the impact.
Answer: Dear reviewer. Thanks for your contributions, your comments rally improved the article and all the team learn a lot.
The objective and the results were changed using concomitance and not impact for a more appropriate information for the readers.
"For this reason, the aim of the present study was to analyze the concomitance of SS in the HRQoL of Spanish children and adolescents and to estimate the prevalence of the different features of SS".
"Table 3 summarizes the data on the different PRO in actual results and estimates (with 95% CI). The results of the EuroQoL-5D are presented as percentage of children and adolescents in each of the levels of concomitance (from lower to higher impact), as the calculated score, and as the VAS."
We decided not to include the regression analysis.
The normalized EuroQoL variable and TSK explain 46.8% of the pain on the PEDS scale, but the best option is considering concomitance and not the impact.
Finally, you write: "Despite their importance for optimal care and treatment success, [...] patient-reported outcomes (PROs) are essential for obtaining more detailed information about the patient’s health". I'm agree, PROs are essential: why "despite their importance ...."? - I don't understand!
Answer: My sincerest apologies. That introductory expression was intended to be used in order to elaborate further on the matter later on. However, we decided to reword it and the introductory phrase remained there.

Reviewer 3 Report
I would like to thank the authors for the revision. There are only minor comments left:
1)P9L219-22: “Future studies should consider a conservative treatment approach, such as exercise protocols on these patients, with the aim of improving their physical condition and thus their quality of life“
This statement in the conclusions section is not supported by the results. The authors did not assess the impact of treatment protocols on PROMs .
I would recommend transferring this statement to the discussion section and only making conclusions in the conclusion section that are backed up by your results. When making statements about what future studies should perform, these statements should be translated from your results.
2)Please add in your limitation section (P9L203-212) that you did not assess treatments in these patients. In this case, you did not assess how treatments (e.g., conservative treatments) affected PROMs. This might be an important limitation.
Author Response
I would like to thank the authors for the revision. There are only minor comments left:
1)P9L219-22: “Future studies should consider a conservative treatment approach, such as exercise protocols on these patients, with the aim of improving their physical condition and thus their quality of life“
This statement in the conclusions section is not supported by the results. The authors did not assess the impact of treatment protocols on PROMs .
I would recommend transferring this statement to the discussion section and only making conclusions in the conclusion section that are backed up by your results. When making statements about what future studies should perform, these statements should be translated from your results.
Answer: this sentence was added in the discussion P9L210-212.
2)Please add in your limitation section (P9L203-212) that you did not assess treatments in these patients. In this case, you did not assess how treatments (e.g., conservative treatments) affected PROMs. This might be an important limitation.
Answer: we include in the data collected the number of patients who received different treatments. However, because of the extension of the study, we decided not included this information. We have information about physical therapy, surgery, and analgesic treatments. If you considered that this information is important, we can add a new table with this information.